# Visual Head Counts: A Promising Method for Efficient Monitoring of Diamondback Terrapins

**Patricia Levasseur [1],\*, Sean Sterrett [2] and Chris Sutherland [1],\***

[1] Department of Environmental Conservation, University of Massachusetts–Amherst, Amherst, MA 01003, USA

[2] Department of Biology, Monmouth University, West Long Branch, NJ 07764, USA

\* Correspondence: plevassuer@umass.edu (P.L.); csutherland@umass.edu (C.S.)

**Abstract:** Determining the population status of the diamondback terrapin (*Malaclemys terrapin* spp.) is challenging due to their ecology and limitations associated with traditional sampling methods. Visual counting of emergent heads offers a promising, efficient, and non-invasive method for generating abundance estimates of terrapin populations across broader spatial scales than has been achieved using capture–recapture, and can be used to quantify determinants of spatial variation in abundance. We conducted repeated visual head count surveys along the shoreline of Wellfleet Bay in Wellfleet, Massachusetts, and analyzed the count data using a hierarchical modeling framework designed specifically for repeated count data: the N-mixture model. This approach allows for simultaneous modeling of imperfect detection to generate estimates of true terrapin abundance. Detection probability was lowest when temperatures were coldest and when wind speed was highest. Local abundance was on average higher in sheltered sites compared to exposed sites and declined over the course of the sampling season. We demonstrate the utility of pairing visual head counts and N-mixture models as an efficient method for estimating terrapin abundance and show how the approach can be used to identifying environmental factors that influence detectability and distribution.

**Keywords:** abundance; detection; diamondback terrapin; *Malaclemys terrapin*; monitoring; N-mixture; salt marsh; visual-head count

## 1. Introduction

The diamondback terrapin (DBT; *Malaclemys terrapin* spp.) is the only estuarine obligate turtle species in North America [1], and as a result, has a long but fragmented coastal range that extends from Cape Cod, Massachusetts, to the Texas Gulf Coast [2]. DBTs are currently listed as protected or regulated in every range state [3]. Despite their conservation status, population assessments have been limited to very local scales, and as a result, comparable and representative status assessments of this imperiled species are generally lacking [3,4].

As salt marsh specialists, DBTs have specific life history and behavioral traits that determine which monitoring techniques are suitable and how that data can be used. For example, DBTs have highly seasonal phenology in their northern range [5], form mating aggregations [6], and have highly specialized terrestrial nesting habitat requirements. The species is also highly mobile [1,7], with movements that are linked with tide cycles [7], and surface regularly to breathe [3,8]. A variety of methods are used to monitor DBTs, including modified crab traps [9–11], hoop traps [12,13], trammel nets [4,12–14], fyke nets [15,16], seines [14,17,18], and dip nets [10,13]. While each method takes some aspects of DBT ecology into consideration, their success, in terms of consistent, reliable, and scalable population estimates has been variable. For example, in Wellfleet Bay (MA), almost four decades of monitoring using capture mark–recapture (CMR) has resulted in over 3000 marked

individuals [19], but failed to produce reliable estimates of population sizes due to low detection rates (see also: [11,13,18]), mature female-biased captures (see also: [9,20]), and variable and opportunistic search effort. These challenges often result in extensive sampling effort being concentrated in extremely localized study areas that are not representative of the landscape [18,20,21].

A promising, but vastly under-utilized, monitoring method for the DBT is visual counting of emergent heads (hereafter, visual head counts), which offers an efficient, non-invasive method for generating abundance estimates of local populations [22]. Because DBTs are the only turtle species to inhabit coastal estuarine habitats, must surface to breathe air, and perform seasonal staging behavior where both sexes congregate to initiate courtship and mating, their biology and behavior lend themselves naturally to methods that rely on detection, but that does not require individual recognition. For example, Isdell et al. [22] used the detection and nondetection of emergent heads to investigate factors that influence site occupancy. Although there is some support for extending the presence–absence surveys to include the counts of heads to estimate relative abundance [4,9], the concept has yet to gain traction. This is despite the fact that visual (point) count surveying is a widely adopted monitoring methodology in ecology [23] with well-established statistical models for analyzing these data (N-mixture model [24]), and both can easily be modified for DBT.

In this study, we conducted visual head count surveys at 38 locations throughout Wellfleet Bay, Massachusetts. Using efficient spatially- and temporally-replicated visual head count surveys and well-established statistical models, we were able to produce estimates of local population size, including linking abundance to shoreline exposure and seasonality, and estimates of how environmental conditions (wind and air temperature) influence detectability. Our results suggest that visual head count surveys are a promising method for monitoring of diamondback terrapins.

## 2. Materials and Methods

### 2.1. Study Area

This study is focused on approximately 50 km of shoreline around Wellfleet Bay (WB), a protected area located in the town of Wellfleet, within Cape Cod Bay, Massachusetts, USA (Figure 1). Wellfleet Bay is a marsh dominated system comprised of many creeks and inlets with an extensive intertidal zone that can exceed 3 meters during spring high tides.

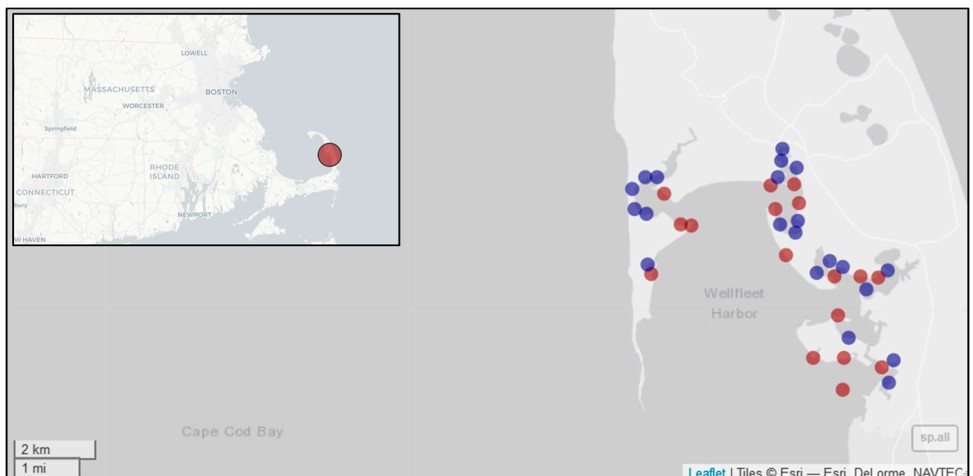

**Figure 1.** The Wellfleet Bay study area. Circles in the main figure show the location of the 38 visual head count surveys (Red: exposed, Blue: sheltered), while the inset shows the geographic location of Wellfleet Bay in the greater Cape Cod Bay, Massachusetts.

We conducted visual head count surveys (visual surveys from here) at 38 locations along the shoreline of WB. Sites were selected using the following approach: First, points were generated every

500 meters along the entire shoreline of WB using the *Generate Points Along Lines* tool in ArcMap 10.6 (ESRI 2018). We used 500 m between points to ensure that on any given day, we would avoid double counting of individuals (i.e., to ensure independence among the sampling locations). Next, we removed points that were located in non-habitat, leaving 44 potential survey sites. We note that 'non-habitat' was defined as areas with no marsh habitat, and thus, our surveys were focused on areas where DBTs would, in theory, be expected to occupy at some point during the tide cycle. Upon initial visits to these 44 sites, six were deemed either inaccessible or unsuitable for surveying (e.g., not enough open water visible to detect surfaced heads), leaving the final 38 suitable sites (Figure 1).

*2.2. Visual Head Count Surveys*

We drove to the sampling locations and accessed each site from land using a handheld GPS unit (Garmin GPSMAP 78, Olathe, Kansas). Visual surveys were conducted by scanning the water from shoreline to shoreline using binoculars, and recording the number of DBT heads that were observed inside a 100 m radius from the survey point. During each site visit, we conducted five (5) such scans, each lasting no longer than 2 min, with a 1-minute break between the end of one scan and the beginning of the next. Surveys were conducted at high tide, when DBTs are most active [7] and demonstrate regular emergence–submergence behavior [8]. Separating scans by one minute was assumed to be sufficient to allow for turnover, and thus variation, in which individuals emerged and were available for detection (Figure A1). Each site was visited at least once each month from May through August 2018 (median number of site visits: 4, range: 3–13). Thus, the data generated from each site visit are five (5) imperfect counts of a population assumed to be constant during the period of counting, but that can vary between site visits and between sites. In total, there were 184 five-scan head count surveys conducted at 38 sites (i.e., 184 unique site-visit combinations).

The area sampled was approximately a 100 m radius semi-circle around the sampling location from the shoreline, extending into the water. This area was identified using a rangefinder (Halo, XL450-7, Grand Prairie). Rangefinders cannot reliably or efficiently detect DBT heads, which are too small, and therefore, were not used to count heads. Instead, proficiency in observer distance estimation was achieved through extensive self-calibration prior to, and regularly during, the sampling season by comparing estimated distances with rangefinder distances of objects easily detected by rangefinders in the water (e.g., boats, buoys).

*2.3. Statistical Analysis*

A natural analytical framework for analyzing repeated counts of a closed population is the N-mixture model [24]. Formally, the counts $y_{ik}$, which are the number of heads observed in scan $k$, where $k = 1, \ldots, 5$, from site $i$, where $i = 1, \ldots, 184$ unique site-by-visit surveys, are assumed to be binomial random variables with a trial size of $N_i$, i.e., the true population size at site $i$, and success probability, $p_{ik}$, which is the probability of detecting an individual in the population at site $i$ during scan $k$. This can be written as follows:

$$y_{ik} \sim \text{Bin}(N_i, p_{ik}). \tag{1}$$

The N-mixture model assumes that individuals are equally detectable, but does allow detectability to be modelled using scan- or site-specific covariates that are assumed to influence detectability. In our case, we considered four environmental covariates that we hypothesized would influence our ability to detect DBT heads: air temperature (Celsius, 'Temp'), cloud cover (clear, <50%, ~50%, >50%, or Overcast, 'Cloud'), wind speed (miles per hour, 'Wind'), and exposure classification ('Expo', see below). Temperature, cloud cover, and wind speed were measured immediately prior to conducting

the visual surveys, and the same covariate value was used for each scan at a site during a single visit. Detection probability can be modeled using a logit-linear model as follows:

$$\text{logit}(p_{ik}) = \beta_0 + \beta_{\text{wind}} \times \text{Wind}_{ik} + \beta_{\text{cloud}} \times \text{Cloud}_{ik} + \beta_{\text{temp}} \times \text{Temp}_{ik} + \beta_{\text{expo}} \times \text{Expo}_{ik} \tag{2}$$

where the intercept ($\beta_0$) and the coefficients for air temperature ($\beta_{\text{temp}}$), cloud cover categories ($\beta_{\text{cloud}}$), the effect of wind ($\beta_{\text{wind}}$), and exposure class ($\beta_{\text{expo}}$), are parameters to be estimated.

The N-mixture model is a hierarchical model, which means that the detection process ($p_{ik}$ above) can be modeled conditionally on, and independent of, the true abundance at a site $N_i$. [24,25]. This ability to explicitly account for imperfect detection while simultaneously estimating variation in true abundance is what makes these types of observation-state hierarchical models so appealing [26]. Formally, abundance at a site, $N_i$, is described as either a Poisson or negative binomially distributed random variable with expected value $\lambda_i$. Preliminary analysis comparing Poisson and negative binomial formulations of the N-mixture model determined that the negative binomial model was preferred (Table A1). Thus, abundance is described as follows:

$$N_i \sim \text{NB}(\lambda_i). \tag{3}$$

As with detection, variation in abundance can be modeled as a function of site-specific covariates using an appropriate generalized linear model (GLM) [25]. For this pilot study, we were primarily interested in demonstrating that head counts could produce data suitable for analysis using the N-mixture model, and that the model could be used to make biologically meaningful inferences. As such, rather than explore the range of hypothesized drivers of DBT abundance, we instead used a broadly defined 'exposure' site classification ('Exposed', Figure 1), where sites were classified as exposed (i.e., sampling sites were located on a stretch of shoreline that were exposed to open water of the larger bay), or unexposed (i.e., sampling sites were located on a stretch of shoreline that were sheltered to the open water of the larger bay). We used this binary exposure category as a covariate on abundance that we assume broadly captures potential differences in habitat quality. For example, we observe larger areas of salt marsh, and specifically *Spartina alterniflora*, habitat in areas of the bay that are more sheltered and less affected by weather-related turbulence. In addition, we also included an effect of seasonality (day since first survey, 'Day') to capture spatiotemporal dynamics that may result in seasonal changes in abundance variation. For the negative binomial regression, an appropriate GLM is a log-linear model:

$$\log(\lambda_i) = \alpha_0 + \alpha_{\text{exposed}} \times \text{Exposed}_i + \alpha_{\text{day}} \times \text{Day}, \tag{4}$$

where the intercept ($\alpha_0$), which is the expected abundance, on the log scale, for exposed sites, and the coefficient measuring the difference between the expected abundance at exposed and unexposed locations ($\alpha_{\text{exposed}}$), and the effect of relative day of the season ($\alpha_{\text{Day}}$) are parameters to be estimated.

Because we were interested in exploring which covariate effects were most important in explaining both detection (i.e., Wind, Cloud, and Temp) and abundance (i.e., Exposed, and Day), we fit all possible combinations of covariate effects models. For detection, this included a null model (constant detection across all sites), univariate models for each of the three covariates, all possible pairs of covariates, and a model with all three covariates included (eight detection models, Table A1). For abundance, this included a null model (constant abundance across all sites), univariate models for each of the two covariates, an additive model with Exposed and Day, and an interactive Exposed-Day model (five abundance models, Table A1). Thus, in total for the negative binomial N-mixture models, we considered 40 models (Table A1). We treated each of the 184 unique site-visit samples as independent sites, acknowledging that because the system is highly dynamic, not doing so would violate the assumption of closure. We analyzed the data using maximum likelihood using the R package 'unmarked' [27], and were ranked according to AIC values where lowest is best [28].

## 3. Results

A total of 184 head count surveys were conducted at 38 spatially distinct sites, each of which was visited at least once each month from May through August 2018 (median number of site visits: 4, range: 3–13). Of the 38 sites, 17 were categorized as exposed, and 21 were categorized as unexposed (Figure 1). Twenty-nine percent (29%) of surveys were conducted under clear skies, 28% conducted in <50% cloud clover, 5% in 50% cloud cover, 20% in >50% and 17% in overcast conditions. Surveys were conducted in wind speeds that ranged from 1 mph to 16 mph. The mean head count in a single scan was 2.65 (median = 0) and ranged from 0 to 91 individuals. DBTs were detected at 36 out of the 38 sites surveyed.

Based on model evaluation using Akaike's Information Criterion (AIC [28]), the best-supported model allowed detection to vary as a function of wind speed, air temperature, and exposure category, and abundance to vary by exposure category and day of the year (i.e., this model had the lowest AIC, Table 1). A model that included cloud cover had some support based on AIC ($\Delta$AIC = 0.12, Table 1), but following recommendations in [29], this term can be considered non-informative because the additional covariate did not improve the support relative to the top model which was simpler by one term. Therefore, below, we report our findings based on the top model: $p$(Temp + Wind + Exposed) $\lambda$(Day × Exposed).

**Table 1.** Model selection table for the negative binomial parameterization of the N-mixture model. Here we show the subset of 10 models that accounted for all the AIC support ($\omega$AIC ≥ 0.01). The models are ranked according to their AIC score (lowest is better). The Detection ($p$) and Abundance ($\lambda$) model formulations are provided, as is the number of parameters in the model (K), the AIC score, the difference in AIC relative to the top model ($\Delta$AIC), the AIC weight ($\omega$AIC) which is a measure of relative model support, and the cumulative AIC weights ($\Sigma\omega$AIC).

| Detection | Abundance | K | AIC | $\Delta$AIC | $\omega$AIC | $\Sigma\omega$AIC |
|---|---|---|---|---|---|---|
| $p$(wind + airtemp + expo) | $\lambda$(relday × expo) | 9 | 2442.84 | 0.00 | 0.33 | 0.33 |
| $p$(wind + airtemp + expo) | $\lambda$(relday) | 7 | 2443.66 | 0.81 | 0.22 | 0.56 |
| $p$(wind + airtemp + expo) | $\lambda$(relday + expo) | 8 | 2444.29 | 1.44 | 0.16 | 0.72 |
| $p$(wind + ccov + airtemp + expo) | $\lambda$(relday × expo) | 13 | 2444.85 | 2.00 | 0.12 | 0.84 |
| $p$(wind + ccov + airtemp + expo) | $\lambda$(relday) | 11 | 2447.12 | 4.28 | 0.04 | 0.88 |
| $p$(wind + airtemp) | $\lambda$(relday × expo) | 8 | 2447.34 | 4.50 | 0.04 | 0.92 |
| $p$(wind + ccov + airtemp) | $\lambda$(relday × expo) | 12 | 2447.66 | 4.81 | 0.03 | 0.95 |
| $p$(wind + ccov + airtemp + expo) | $\lambda$(relday + expo) | 12 | 2447.90 | 5.05 | 0.03 | 0.98 |
| $p$(wind + airtemp) | $\lambda$(relday + expo) | 7 | 2449.37 | 6.53 | 0.01 | 0.99 |
| $p$(wind + ccov + airtemp) | $\lambda$(relday + expo) | 11 | 2450.90 | 8.05 | 0.01 | 0.99 |

Detection probability was higher in unexposed sites ($\beta_{expo}$ = 1.70, 95% CI: 2.39–1.01), and lowest when surveys were conducted in cold temperatures during high winds (Figure 2). Detection probability was negatively influenced by wind ($\beta_{wind}$ = −0.19, 95% CI: −0.27–−0.11, Figure 2) and positively influenced by air temperature ($\beta_{temp}$ = 0.18, 95% CI: 0.12–0.25, Figure 2). For example, a survey conducted in the warmest observed temperature (32.2 C) and lowest wind (1 mph) would have an expected detection probability of 0.22 at an exposed site (95% CI: 0.10–0.43, Figure 2), and 0.61 at an unexposed site (95% CI: 0.42–0.77, Figure 2). Conversely, in the coldest temperatures (11.1 °C) and highest winds (16 mph), DBT would be practically undetectable at both exposed and unexposed sites (0.0003, 95% CI: 0.0001–0.001 and 0.002, 95% CI: 0.0007–0.005, respectively, Figure 2). Thus, maximum detection probability is achieved when sampling in warm conditions and low winds.

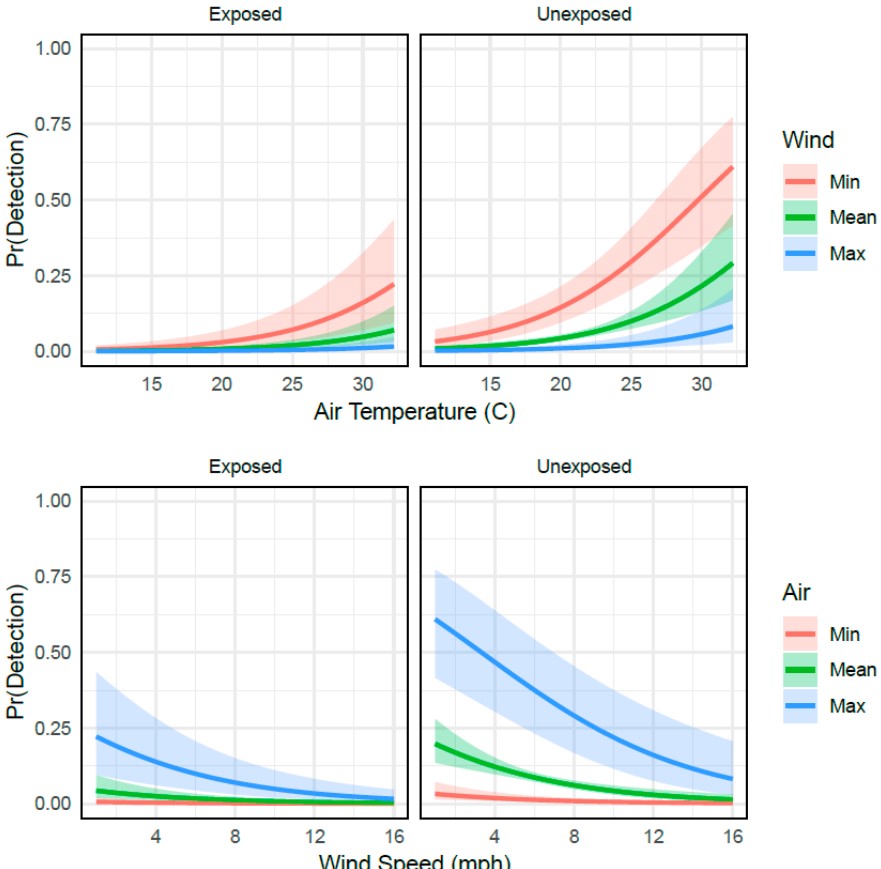

**Figure 2.** Per individual detection probability of diamondback terrapin heads by exposure category (Exposed, *left*; Unexposed, *right*) and as a function of air temperature (Celsius, *top*) and wind speed (miles per hour, *bottom*). The influence of air temperature on detectability is shown at the minimum, mean, and maximum wind speed observed during surveys (*top*). The influence of wind speed on detectability is shown at the minimum, mean, and maximum air temperature observed during surveys (*bottom*). Solid lines are maximum likelihood estimates, and polygons are the 95% confidence intervals.

Based on AIC, the top abundance model included the exposure classification, day of the year, and an interaction between the two (Table 1). Unexposed sites had, on average, higher expected abundance than exposed sites at the beginning of the season ($\alpha_{\text{unexposed}}$ = 0.24, 95% CI: −1.05–1.54, estimates on the log scale). Abundance declined through the season, although this decline was faster in unexposed sites: $\alpha_{\text{relday:unexposed}}$ = −0.010 (95% CI: −0.027–0.007) than exposed sites: $\alpha_{\text{relday:exposed}}$ = 0.030 (95% CI: −0.039−−0.018, Figure 3). The expected abundance at exposed sites on the first day of the sampling season (day$_0$: 11 May 2018) is 125 terrapins (95% CI: 39–406) and on the last day (day$_{109}$: 28 August 2018) was 42 (95% CI: 12–144). In contrast, expected abundance at unexposed sites ranged from 160 (95% CI: 84–305) at the start of the season to 7 (95% CI: 3–15) at the end.

Finally, we assessed the goodness of fit of our AIC-top model using parametric bootstrapping [25,30]. We conducted 1000 parametric bootstrap simulations (i.e., simulated data from our fitted model) and computed two commonly used goodness of fit statistics, the sums of squares (SSE, the sum of the squared residuals, [25]) and the Freeman Tukey statistic (a metric that compares observations to expectations under the model [25]). For both goodness of fit statistics, the observed statistic (SSE or Freeman Tukey) did not differ significantly from those produced via bootstrap simulations ($p_{\text{SSE}}$ = 0.153 and $p_{\text{FT}}$ = 0.215) suggesting that our model is a good fit to the observed data (Table A2, Figure 4).

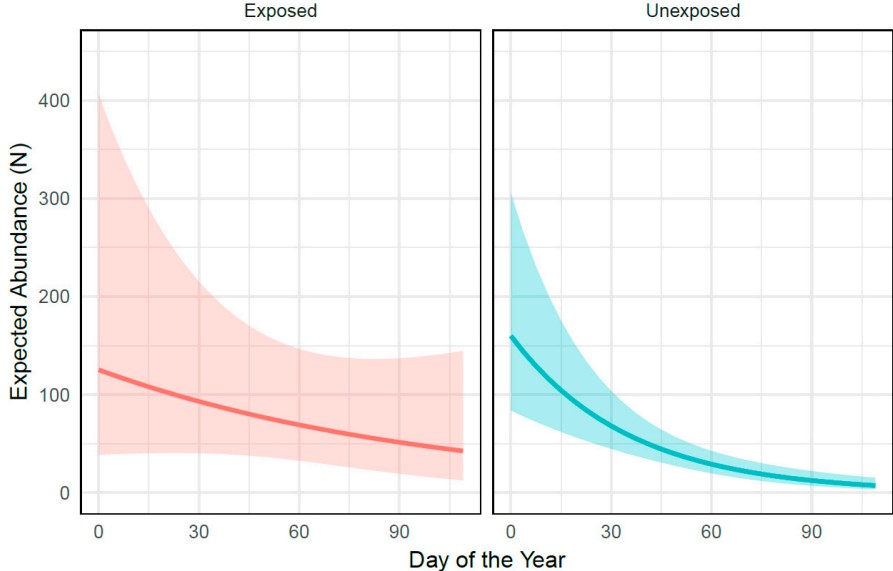

**Figure 3.** Model estimates of expected abundance ($\lambda_{exposed/unexposed}$) as a function the number of days since the first survey (day$_0$: 11 May 2018) separated out by exposure category (*left*: exposed, *right*: unexposed). Bold lines are maximum likelihood estimates, and shaded polygons are the 95% confidence intervals.

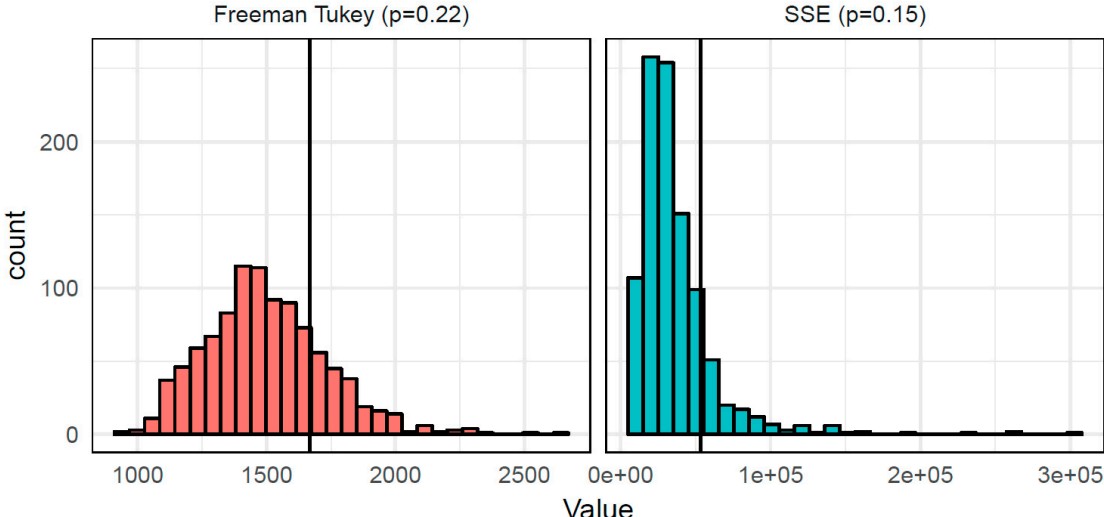

**Figure 4.** Visualization of the goodness of fit test results for the AIC-top model. The histograms show the bootstrapped (simulated) test statistics with the associated *p*-value in the title (*left*: Freeman Tukey, *right*: SSE). The vertical black line shows the observed test statistic and demonstrates that it is a reasonable assumption that the data could arise from the model used for inference.

## 4. Discussion

In this study, we demonstrate how pairing visual head count surveys with N-mixture modeling offers a complementary data collection and analysis framework for efficiently estimating local diamondback terrapin population sizes, while simultaneously accounting for factors that affect detectability and spatial variation in abundance. We found that detection was highest in warmer conditions and, as expected, negatively influenced by wind speed, and that DBTs were more likely to be detected in unexposed sites. Our results suggest that abundance was higher at sites along sheltered stretches of shoreline relative to sites that were exposed, and that per site abundance reduced over the course of the season, a reduction that was more pronounced at unexposed sites. We argue that

visual head count surveys are indeed a promising and efficient method of estimating abundance for this important estuarine obligate turtle.

Studies of DBTs have been largely focused on capture mark–recapture (CMR) and are plagued with reported issues of detectability [9,18,19]. Our study confirms that this challenge is not restricted to CMR, but also impacts visual head counts (see also [22]). However, repeat-survey designs, such as those called for under the N-mixture model, are developed specifically to capture variability in imperfect counts and relate that variability to variation in factors (e.g., environmental) thought to influence detectability. While robust design CMR is designed in the same way, an important distinction is that visual head counts do not require physical capture and are designed in line with the species surfacing and aggregation behavior to maximize detection, and this appears to yield far greater sample sizes. For example, Isdell et al. [22] were able to relate variation in DBT site occupancy to terrestrial–aquatic connectivity using presence–absence surveys of emerged heads; our approach extended this protocol to explicitly model the counts to make inferences about abundance rather than occupancy. While this precludes explicit estimation of demographic rates (survival and fecundity), when the inference objective is to estimate occurrence or abundance patterns then estimates from unmarked individuals as we have presented here have obvious value (see Figure 3).

Applying the N-mixture model to repeated head count surveys, we were able to both identify, and correct for, factors that influenced detectability and therefore, estimate population size free of these specific biases. Specifically, for DBT in Wellfleet Bay, wind reduced detectability, detection was highest in warm conditions, and terrapins were more likely to be detected at sheltered sites than exposed sites (Figure 2). The negative effect of wind on detection was expected, and likely related to either increase wave chop, making heads more difficult to observe or, behavior-related, that DBTs surface less in higher winds. Likewise, the positive effect of air temperature on detectability is intuitive, considering DBTs are ectothermic and are most active at high tides when our surveys are conducted [7]. Detectability was higher at unexposed sites (Figure 2), likely due to the fact that they are relatively (compared to exposed sites) sheltered from the weather and tidal systems that can reduce detectability. Linking potential behavioral responses to environmental conditions using hierarchical models as we have done here, and as demonstrated by Isdell et al. [22], has potential implications for other capture methods that rely on visual detection of terrapins (e.g., dip netting, drones). Our study suggests that the ideal conditions for conducting visual head count surveys in this system are in warm conditions with low-to-no wind.

One of the most appealing features of combining visual head counts and N-mixture modeling is the ability to generate estimates of local (scan area) population size for several locations within Wellfleet bay, and link those estimates to spatially varying covariates. This is in contrast with intensive CMR efforts, which often require multiple seasons to generate sufficient recaptures [18,19], and have been restricted to just two locations within the bay [19]. This is common throughout the range where CMR estimates of abundance are typically, and justifiably, made over a spatially restricted area that is small, and thus not representative, of the larger population of interest [11,12]. Moreover, this expansion of the spatial coverage and the statistical inference about DBT population status, at least in Wellfleet Bay, is achieved far more efficiently than traditional methods [9,15], and arguably yields more management-relevant results. For example, the ability to move beyond point estimates of abundance and start to relate spatial variation in local abundance to habitat characteristics or environmental conditions is potentially far more valuable information to inform proactive species- and habitat-specific conservation action.

Our focus here was to demonstrate the utility of visual head counts and N-mixture models as an efficient method for estimating abundance, and as such, we did not exhaustively explore all potential hypothesized predictors of abundance. Instead, we focused on differences between sheltered and exposed shorelines as a simple catch-all proxy for a wide range of environmental disturbance, and importantly, the extent and distribution of preferred salt marsh habitat. However, we were able to capture interesting and intuitive spatiotemporal patterns. We found that abundance was higher in

unexposed sites at the beginning of the sampling season, but was lower than exposed sites by the end of the season due to a steeper population decline over the season. Despite being a crude measure of habitat quality, these model predictions are in line with what we would expect: unexposed sites experience less environmental disturbance (e.g., turbidity) where intact saltmarsh habitat is more likely to be found. The start of our sampling period coincides with post-brumation emergence of the DBT, and if sheltered areas are indeed a proxy for higher quality habitat, then our prediction of higher abundance in sheltered areas early in the season would be consistent with the idea that these areas are likely locations for courting and mating aggregations [6,7]. Likewise, DBT disaggregation, and in particular terrestrial nesting by females [7], is consistent with our prediction of reduced abundances in the better quality areas, coupled with an overall reduction in abundance. Our predictions do, however, have large associated uncertainty, especially for exposed sites (Figure 3), and should be interpreted with caution. While our measure of exposure broadly captures abundance patterns in Wellfleet Bay, there is more to be done to characterize finer scale drivers of spatiotemporal variation before these results can be used to inform conservation action; this remains an important area of research throughout the DBT range [3]. Encouragingly, though, we have demonstrated that the coupled field and statistical framework we have described here can be used to achieve this, but requires identifying the appropriate data that link covariates to standing hypotheses about DBT ecology. Our next step is to identify and develop a suite of spatiotemporal covariates to formally test hypotheses about drivers of population sizes in both space and time.

## 5. Conclusions

The visual head count methodology we describe naturally matches the ecology of DBT, is easy to conduct and require very little training (i.e., low intensive), and, as demonstrated here, can be used to generate meaningful spatially referenced estimates of local abundance across a region of interest. This contrasts substantially with the widely applied CMR approaches, which involve substantial effort, both in terms of time and expertise (i.e., highly intensive), and often suffer from extremely low capture rates that require multiple sampling seasons to generate abundance estimates, and as a result, are typically limited in terms of spatial coverage. Further, we demonstrate the application of N-mixture models, the canonical analytical framework for analyzing such repeated count data, and were able to identify ideal survey conditions that will maximize detection in future surveys, while revealing site-specific variation in estimates of abundance that are consistent with habitat preferences and phenology. We propose visual head count surveys and associated hierarchical modeling framework, as a promising method for generating spatially explicit estimates of diamondback terrapin abundance and investigating drivers of spatiotemporal variation in abundance.

**Author Contributions:** Conceptualization, P.L. and C.S.; methodology, P.L., C.S., S.S.; formal analysis, P.L., C.S.; investigation, P.L.; writing—original draft preparation, P.L.; writing—review and editing, P.L., C.S., S.S.; visualization, P.L., C.S.; supervision, C.S.; project administration, C.S., P.L.; funding acquisition, C.S., P.L.

**Funding:** This research was funded by Massachusetts Division of Fisheries and Wildlife Natural Heritage and Endangered Species Program MESA mitigation funds.

**Acknowledgments:** We would like to thank Mike Jones (Mass Wildlife NHESP), Bob Prescott and Mark Faherty (Mass Audubon Wellfleet Bay Wildlife Sanctuary) and Rachel Katz (USFWS) for their valuable input.

**Conflicts of Interest:** The authors declare no conflict of interest.

# Appendix A

**Table A1.** Full model selection table for the negative binomial (left side) and Poisson parameterizations of the N-mixture model. Here we show all 40 models for each parameterization. The models are ranked according to their negative binomial AIC scores (i.e., the AIC-preferred model, lowest is better). The Detection ($p$) and Abundance ($\lambda$) model formulations are provided, as is the number of parameters in the model (K), the AIC score, the difference in AIC relative to the top model ($\Delta$AIC), the AIC weight ($\omega$AIC) which is a measure of relative model support, and the cumulative AIC weights ($\Sigma\omega$AIC). The '*' denotes interactions between terms and ($\cdot$) denotes the intercept only (or null) model.

| Detection | Abundance | Negative Binomial | | | | | Poisson | | | |
|---|---|---|---|---|---|---|---|---|---|---|
| | | K | AIC | $\Delta$AIC | $\omega$AIC | $\Sigma\omega$AIC | K | AIC | $\Delta$AIC | $\omega$AIC | $\Sigma\omega$AIC |
| $p$(wind + airtemp + expo) | $\lambda$(relday × expo) | 9 | 2442.84 | 0.00 | 0.33 | 0.33 | 8 | 4257.74 | 1814.89 | 0.00 | 1.00 |
| $p$(wind + airtemp + expo) | $\lambda$(relday) | 7 | 2443.66 | 0.81 | 0.22 | 0.56 | 6 | 4294.18 | 1851.33 | 0.00 | 1.00 |
| $p$(wind + airtemp + expo) | $\lambda$(relday + expo) | 8 | 2444.29 | 1.44 | 0.16 | 0.72 | 7 | 4272.59 | 1829.75 | 0.00 | 1.00 |
| $p$(wind + ccov + airtemp + expo) | $\lambda$(relday × expo) | 13 | 2444.85 | 2.00 | 0.12 | 0.84 | 12 | 4200.42 | 1757.57 | 0.00 | 1.00 |
| $p$(wind + ccov + airtemp + expo) | $\lambda$(relday) | 11 | 2447.12 | 4.28 | 0.04 | 0.88 | 10 | 4257.23 | 1814.39 | 0.00 | 1.00 |
| $p$(wind + airtemp) | $\lambda$(relday × expo) | 8 | 2447.34 | 4.50 | 0.04 | 0.92 | 7 | 4314.40 | 1871.55 | 0.00 | 1.00 |
| $p$(wind + ccov + airtemp) | $\lambda$(relday × expo) | 12 | 2447.66 | 4.81 | 0.03 | 0.95 | 11 | 4222.54 | 1779.70 | 0.00 | 1.00 |
| $p$(wind + ccov + airtemp + expo) | $\lambda$(relday + expo) | 12 | 2447.90 | 5.05 | 0.03 | 0.98 | 11 | 4227.21 | 1784.36 | 0.00 | 1.00 |
| $p$(wind + airtemp) | $\lambda$(relday + expo) | 7 | 2449.37 | 6.53 | 0.01 | 0.99 | 6 | 4346.61 | 1903.77 | 0.00 | 1.00 |
| $p$(wind + ccov + airtemp) | $\lambda$(relday + expo) | 11 | 2450.90 | 8.05 | 0.01 | 0.99 | 10 | 4265.13 | 1822.28 | 0.00 | 1.00 |
| $p$(wind + ccov + expo) | $\lambda$(relday × expo) | 12 | 2454.49 | 11.65 | 0.00 | 1.00 | 11 | 4231.32 | 1788.48 | 0.00 | 1.00 |
| $p$(wind + ccov) | $\lambda$(relday × expo) | 11 | 2454.71 | 11.87 | 0.00 | 1.00 | 10 | 4240.50 | 1797.66 | 0.00 | 1.00 |
| $p$(wind + ccov + airtemp) | $\lambda$(relday) | 10 | 2454.88 | 12.03 | 0.00 | 1.00 | 9 | 4425.24 | 1982.39 | 0.00 | 1.00 |
| $p$(airtemp + expo) | $\lambda$(relday × expo) | 8 | 2456.70 | 13.86 | 0.00 | 1.00 | 7 | 4326.87 | 1884.03 | 0.00 | 1.00 |
| $p$(wind + ccov + expo) | $\lambda$(relday) | 10 | 2456.70 | 13.86 | 0.00 | 1.00 | 9 | 4352.01 | 1909.16 | 0.00 | 1.00 |
| $p$(wind + airtemp) | $\lambda$(relday) | 6 | 2456.77 | 13.92 | 0.00 | 1.00 | 5 | 4506.19 | 2063.34 | 0.00 | 1.00 |
| $p$(ccov + airtemp + expo) | $\lambda$(relday × expo) | 12 | 2457.24 | 14.40 | 0.00 | 1.00 | 11 | 4259.03 | 1816.18 | 0.00 | 1.00 |
| $p$(airtemp + expo) | $\lambda$(relday) | 6 | 2457.40 | 14.55 | 0.00 | 1.00 | 5 | 4374.93 | 1932.09 | 0.00 | 1.00 |
| $p$(airtemp + expo) | $\lambda$(relday + expo) | 7 | 2457.67 | 14.83 | 0.00 | 1.00 | 6 | 4344.72 | 1901.87 | 0.00 | 1.00 |
| $p$(wind + ccov + expo) | $\lambda$(relday + expo) | 11 | 2458.35 | 15.51 | 0.00 | 1.00 | 10 | 4269.38 | 1826.54 | 0.00 | 1.00 |
| $p$(wind + ccov) | $\lambda$(relday + expo) | 10 | 2458.36 | 15.51 | 0.00 | 1.00 | 9 | 4278.68 | 1835.83 | 0.00 | 1.00 |
| $p$(wind + airtemp + expo) | $\lambda$($\cdot$) | 6 | 2458.46 | 15.61 | 0.00 | 1.00 | 5 | 4483.82 | 2040.97 | 0.00 | 1.00 |
| $p$(ccov + airtemp + expo) | $\lambda$(relday) | 10 | 2458.69 | 15.85 | 0.00 | 1.00 | 9 | 4326.81 | 1883.97 | 0.00 | 1.00 |
| $p$(ccov + airtemp + expo) | $\lambda$(relday + expo) | 11 | 2459.79 | 16.94 | 0.00 | 1.00 | 10 | 4285.82 | 1842.98 | 0.00 | 1.00 |
| $p$(wind) | $\lambda$(relday × expo) | 7 | 2460.12 | 17.28 | 0.00 | 1.00 | 6 | 4369.22 | 1926.37 | 0.00 | 1.00 |
| $p$(wind + airtemp + expo) | $\lambda$(expo) | 7 | 2460.28 | 17.44 | 0.00 | 1.00 | 6 | 4428.94 | 1986.09 | 0.00 | 1.00 |
| $p$(wind + expo) | $\lambda$(relday × expo) | 8 | 2460.46 | 17.62 | 0.00 | 1.00 | 7 | 4348.12 | 1905.28 | 0.00 | 1.00 |

**Table A1.** *Cont.*

| Detection | Abundance | Negative Binomial | | | | | Poisson | | | | |
|---|---|---|---|---|---|---|---|---|---|---|---|
| | | K | AIC | ΔAIC | ωAIC | ΣωAIC | K | AIC | ΔAIC | ωAIC | ΣωAIC |
| $p$(ccov + airtemp) | $\lambda$(relday × expo) | 11 | 2460.48 | 17.63 | 0.00 | 1.00 | 10 | 4295.68 | 1852.83 | 0.00 | 1.00 |
| $p$(wind + ccov + airtemp + expo) | $\lambda$(·) | 10 | 2461.39 | 18.55 | 0.00 | 1.00 | 9 | 4435.15 | 1992.31 | 0.00 | 1.00 |
| $p$(wind + ccov) | $\lambda$(relday) | 9 | 2461.40 | 18.56 | 0.00 | 1.00 | 8 | 4433.07 | 1990.22 | 0.00 | 1.00 |
| $p$(wind + expo) | $\lambda$(relday) | 6 | 2461.52 | 18.68 | 0.00 | 1.00 | 5 | 4462.58 | 2019.74 | 0.00 | 1.00 |
| $p$(wind + ccov + expo) | $\lambda$(·) | 9 | 2462.22 | 19.37 | 0.00 | 1.00 | 8 | 4444.15 | 2001.31 | 0.00 | 1.00 |
| $p$(wind + airtemp) | $\lambda$(expo) | 6 | 2462.84 | 19.99 | 0.00 | 1.00 | 5 | 4476.29 | 2033.45 | 0.00 | 1.00 |
| $p$(ccov + airtemp) | $\lambda$(relday + expo) | 10 | 2463.17 | 20.32 | 0.00 | 1.00 | 9 | 4338.50 | 1895.65 | 0.00 | 1.00 |
| $p$(wind) | $\lambda$(relday + expo) | 6 | 2463.20 | 20.35 | 0.00 | 1.00 | 5 | 4397.94 | 1955.09 | 0.00 | 1.00 |
| $p$(wind + ccov + airtemp + expo) | $\lambda$(expo) | 11 | 2463.21 | 20.36 | 0.00 | 1.00 | 10 | 4368.54 | 1925.70 | 0.00 | 1.00 |
| $p$(wind + expo) | $\lambda$(relday + expo) | 7 | 2463.51 | 20.66 | 0.00 | 1.00 | 6 | 4376.51 | 1933.67 | 0.00 | 1.00 |
| $p$(wind + ccov) | $\lambda$(expo) | 9 | 2464.18 | 21.34 | 0.00 | 1.00 | 8 | 4382.69 | 1939.85 | 0.00 | 1.00 |
| $p$(wind + ccov + expo) | $\lambda$(expo) | 10 | 2464.21 | 21.36 | 0.00 | 1.00 | 9 | 4367.07 | 1924.22 | 0.00 | 1.00 |
| $p$(airtemp) | $\lambda$(relday x expo) | 7 | 2464.22 | 21.38 | 0.00 | 1.00 | 6 | 4411.12 | 1968.28 | 0.00 | 1.00 |
| $p$(wind + ccov + airtemp) | $\lambda$(expo) | 10 | 2464.36 | 21.52 | 0.00 | 1.00 | 9 | 4383.33 | 1940.49 | 0.00 | 1.00 |
| $p$(wind + expo) | $\lambda$(·) | 5 | 2465.54 | 22.70 | 0.00 | 1.00 | 4 | 4542.02 | 2099.18 | 0.00 | 1.00 |
| $p$(airtemp) | $\lambda$(relday + expo) | 6 | 2465.99 | 23.15 | 0.00 | 1.00 | 5 | 4447.64 | 2004.79 | 0.00 | 1.00 |
| $p$(wind) | $\lambda$(expo) | 5 | 2466.74 | 23.89 | 0.00 | 1.00 | 4 | 4478.46 | 2035.61 | 0.00 | 1.00 |
| $p$(wind + expo) | $\lambda$(expo) | 6 | 2466.99 | 24.14 | 0.00 | 1.00 | 5 | 4452.53 | 2009.69 | 0.00 | 1.00 |
| $p$(airtemp + expo) | $\lambda$(·) | 5 | 2469.20 | 26.35 | 0.00 | 1.00 | 4 | 4538.96 | 2096.12 | 0.00 | 1.00 |
| $p$(wind) | $\lambda$(relday) | 5 | 2469.22 | 26.38 | 0.00 | 1.00 | 4 | 4556.42 | 2113.57 | 0.00 | 1.00 |
| $p$(ccov + airtemp) | $\lambda$(relday) | 9 | 2469.49 | 26.64 | 0.00 | 1.00 | 8 | 4517.83 | 2074.98 | 0.00 | 1.00 |
| $p$(ccov + expo) | $\lambda$(relday × expo) | 11 | 2469.57 | 26.73 | 0.00 | 1.00 | 10 | 4319.56 | 1876.71 | 0.00 | 1.00 |
| $p$(ccov) | $\lambda$(relday × expo) | 10 | 2470.03 | 27.18 | 0.00 | 1.00 | 9 | 4341.84 | 1899.00 | 0.00 | 1.00 |
| $p$(airtemp + expo) | $\lambda$(expo) | 6 | 2470.97 | 28.12 | 0.00 | 1.00 | 5 | 4477.94 | 2035.10 | 0.00 | 1.00 |
| $p$(ccov + expo) | $\lambda$(relday) | 9 | 2471.48 | 28.64 | 0.00 | 1.00 | 8 | 4458.38 | 2015.53 | 0.00 | 1.00 |
| $p$(wind + ccov) | $\lambda$(·) | 8 | 2472.54 | 29.69 | 0.00 | 1.00 | 7 | 4539.74 | 2096.90 | 0.00 | 1.00 |
| $p$(ccov + airtemp + expo) | $\lambda$(·) | 9 | 2473.24 | 30.40 | 0.00 | 1.00 | 8 | 4502.02 | 2059.18 | 0.00 | 1.00 |
| $p$(expo) | $\lambda$(relday × expo) | 7 | 2473.30 | 30.46 | 0.00 | 1.00 | 6 | 4409.04 | 1966.20 | 0.00 | 1.00 |
| $p$(ccov + expo) | $\lambda$(relday + expo) | 10 | 2473.47 | 30.63 | 0.00 | 1.00 | 9 | 4358.68 | 1915.83 | 0.00 | 1.00 |
| $p$(ccov) | $\lambda$(relday + expo) | 9 | 2473.60 | 30.76 | 0.00 | 1.00 | 8 | 4381.27 | 1938.43 | 0.00 | 1.00 |
| $p$(expo) | $\lambda$(relday) | 5 | 2473.83 | 30.98 | 0.00 | 1.00 | 4 | 4538.54 | 2095.69 | 0.00 | 1.00 |
| $p$(wind + ccov + airtemp) | $\lambda$(·) | 9 | 2474.03 | 31.19 | 0.00 | 1.00 | 8 | 4540.46 | 2097.62 | 0.00 | 1.00 |
| $p$(·) | $\lambda$(relday × expo) | 6 | 2474.31 | 31.47 | 0.00 | 1.00 | 5 | 4457.35 | 2014.50 | 0.00 | 1.00 |

**Table A1.** *Cont.*

| Detection | Abundance | Negative Binomial | | | | | Poisson | | | | |
|---|---|---|---|---|---|---|---|---|---|---|---|
| | | K | AIC | ΔAIC | ωAIC | ΣωAIC | K | AIC | ΔAIC | ωAIC | ΣωAIC |
| $p$(ccov + airtemp + expo) | $\lambda$(expo) | 10 | 2475.23 | 32.38 | 0.00 | 1.00 | 9 | 4433.41 | 1990.57 | 0.00 | 1.00 |
| $p$(expo) | $\lambda$(relday + expo) | 6 | 2475.77 | 32.93 | 0.00 | 1.00 | 5 | 4439.17 | 1996.33 | 0.00 | 1.00 |
| $p$(airtemp) | $\lambda$(expo) | 5 | 2476.09 | 33.24 | 0.00 | 1.00 | 4 | 4554.15 | 2111.30 | 0.00 | 1.00 |
| $p$(wind + airtemp) | $\lambda$(·) | 5 | 2476.26 | 33.42 | 0.00 | 1.00 | 4 | 4640.62 | 2197.78 | 0.00 | 1.00 |
| $p$(·) | $\lambda$(relday + expo) | 5 | 2476.82 | 33.98 | 0.00 | 1.00 | 4 | 4486.70 | 2043.86 | 0.00 | 1.00 |
| $p$(airtemp) | $\lambda$(relday) | 5 | 2476.85 | 34.00 | 0.00 | 1.00 | 4 | 4629.31 | 2186.46 | 0.00 | 1.00 |
| $p$(ccov + airtemp) | $\lambda$(expo) | 9 | 2476.89 | 34.04 | 0.00 | 1.00 | 8 | 4475.53 | 2032.69 | 0.00 | 1.00 |
| $p$(ccov + expo) | $\lambda$(·) | 8 | 2476.97 | 34.12 | 0.00 | 1.00 | 7 | 4539.94 | 2097.10 | 0.00 | 1.00 |
| $p$(expo) | $\lambda$(·) | 4 | 2477.26 | 34.42 | 0.00 | 1.00 | 3 | 4605.42 | 2162.58 | 0.00 | 1.00 |
| $p$(wind) | $\lambda$(·) | 4 | 2477.41 | 34.56 | 0.00 | 1.00 | 3 | 4644.02 | 2201.18 | 0.00 | 1.00 |
| $p$(ccov + expo) | $\lambda$(expo) | 9 | 2478.54 | 35.70 | 0.00 | 1.00 | 8 | 4447.38 | 2004.54 | 0.00 | 1.00 |
| $p$(expo) | $\lambda$(expo) | 5 | 2478.54 | 35.70 | 0.00 | 1.00 | 4 | 4506.40 | 2063.55 | 0.00 | 1.00 |
| $p$(ccov) | $\lambda$(expo) | 8 | 2478.59 | 35.75 | 0.00 | 1.00 | 7 | 4477.81 | 2034.97 | 0.00 | 1.00 |
| $p$(ccov) | $\lambda$(relday) | 8 | 2479.50 | 36.66 | 0.00 | 1.00 | 7 | 4550.86 | 2108.02 | 0.00 | 1.00 |
| $p$(·) | $\lambda$(expo) | 4 | 2479.64 | 36.79 | 0.00 | 1.00 | 3 | 4558.73 | 2115.88 | 0.00 | 1.00 |
| $p$(·) | $\lambda$(relday) | 4 | 2486.83 | 43.99 | 0.00 | 1.00 | 3 | 4648.64 | 2205.80 | 0.00 | 1.00 |
| $p$(ccov) | $\lambda$(·) | 7 | 2490.58 | 47.74 | 0.00 | 1.00 | 6 | 4647.29 | 2204.45 | 0.00 | 1.00 |
| $p$(ccov + airtemp) | $\lambda$(·) | 8 | 2490.62 | 47.77 | 0.00 | 1.00 | 7 | 4645.78 | 2202.94 | 0.00 | 1.00 |
| $p$(airtemp) | $\lambda$(·) | 4 | 2493.26 | 50.42 | 0.00 | 1.00 | 3 | 4722.14 | 2279.29 | 0.00 | 1.00 |
| $p$(·) | $\lambda$(·) | 3 | 2494.67 | 51.83 | 0.00 | 1.00 | 2 | 4725.64 | 2282.80 | 0.00 | 1.00 |

**Table A2.** Goodness of fit test statistics from a parametric bootstrap of 1000 simulations from the AIC-top model. SSE is the sums of squares and Freeman Tukey compares the observed data to that expected under the model. In the table, $\theta_{obs}$ is the observed test statistic, i.e., that calculated from the actual data, $\theta_{boot}$ is the statistic from data generated from the model, the mean and standard deviation of the differences are the difference from the observed and each simulated data set, and **Pr($\theta$boot > $\theta$obs)** is the probability that the observed statistic is greater than expected as compared to the bootstrapped distribution (i.e., $p < 0.05$ could be interpreted as significantly different). Both p-values are > 0.05, and the model is therefore considered to be adequate.

| Test Statistic | $\theta_{obs}$ | Mean($\theta_{obs} - \theta_{boot}$) | SD($\theta_{obs} - \theta_{boot}$) | Pr($\theta_{boot} > \theta_{obs}$) |
|---|---|---|---|---|
| SSE | 53,435 | 16,737 | 26,883 | 0.153 |
| Freeman Tukey | 1667 | 169 | 236 | 0.215 |

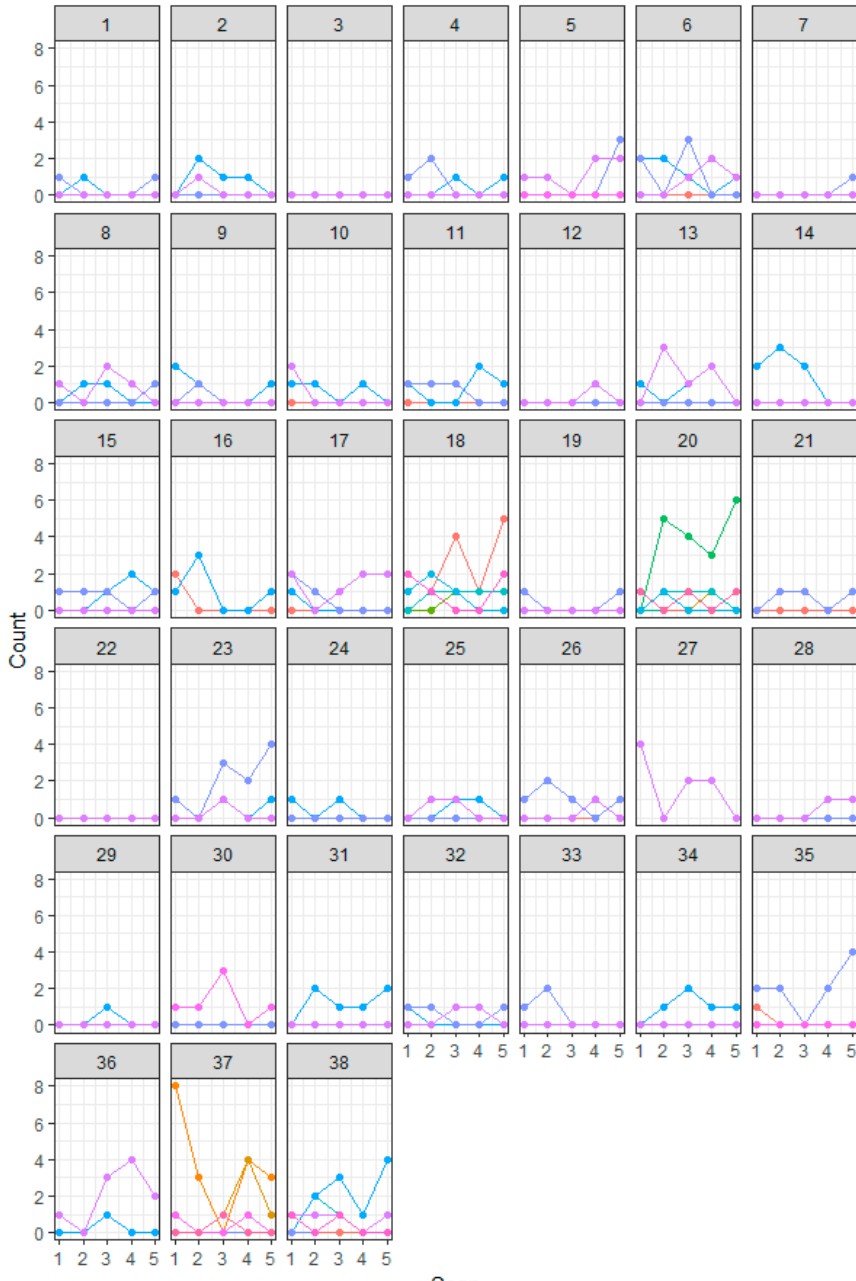

**Figure A1.** Visualization of the within-visit variation in counts at sites with at least one observed 0 which we use to demonstrated that variation in the number of heads counted can be related to turnover of specific individuals at the surface. Each panel represents a site, each line within a panel represents a visit, and the points joined by the lines show the number of heads observed in each of the five scans (blue to red color shades correspond to scan one through five). Here we show only visits that had at least a single count of zero to emphasize the fact that there is a high rate of turnover within visits (demonstrated by the variation between scans). We note that these observations are subject to imperfect detection, but conditions for each visit were similar, and in the absence of available empirical data on emergence rates, this serves as good evidence that we are not simply counting exactly the same individuals in each scan, i.e., there is randomness associated with which individuals are available to be detected from scan-to-scan. Raw data for all data are provided here *archived address here*.

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
