# Peer review of "Visual Head Counts: A Promising Method for Efficient Monitoring of Diamondback Terrapins"

_diversity, doi:10.3390/d11070101_

Round 1

Reviewer 1 Report

This analysis presents a new method for surveying diamond-back terrapins using N-mixture modeling. Overall the paper is straightforward and generally well-written (though with some typos, which I've tried to note below), though I have some overarching concerns.

1)      I would have liked to seen a preliminary analysis just to confirm that the N-mixture model being used can extract correct parameter estimates when applied to simulated data (with known parameter values). By my estimation there are 144 parameters being estimated (138 lambdas and the model coefficients in Equations 2 and 4). The authors have put no hierarchical structure on the lambdas that might aid in estimation. For example, I would assume that adjacent sites would be more similar than distant sites, or maybe groups of sites can be grouped into regions with a single lambda estimated for all the sites in a region. Even a hierarchical model requiring all the lambdas to follow a distribution (perhaps the Normal) would help constrain estimation. I cannot find anywhere that the confidence (or credible) intervals on the lambdas are presented. Figure 4 shows the draws from the mean lambda for each site, but really it should show draws from samples from the lambda estimates. In other words, first you have to draw a lambda value (which incorporates uncertainty in lambda) and then draw from the Poisson, which then incorporates the stochasticity built into the relationships between abundance and density. It very well may be that the uncertainty for lambda is actually larger that the variation from the Poisson, which is all that is shown in Figure 4. Without information on the precision of the lambda estimates, its hard to know how valuable this method is with only 5 replicate survey passes for each site x sample combination.

2)      I am concerned that the five passes of head scans are so close in time that they are not really independent (as claimed on Lines 149-150). I would imagine that terrapins visible in Pass 1 might still be visible in Pass 2 and so each pass doesn’t really represent an independent sample from the Binomial distribution in Equation 1. It would have been far preferable to return in an hour or maybe the next day to do another head scan, allowing for a better “mixing” of terrapins in the landscape and for a genuinely new and independent sample from the population.

3)      I would liked to have seen more discussion about how abundance (at least estimates of lambda) varied spatially and temporally. Are there areas of high terrapin abundance and do your findings accord with prior studies or biological knowledge of the species habitat preferences?

Detailed comments:

Lines 66-67: “and including lining abundance to shoreline exposure…” This sentence doesn’t make sense to me.

Lines 92-93: Awkward phrasing.

Lines 93-94: I suggest “were conducted, by scanning the water from shoreline to shoreline using binoculars…”

Line 113: Your wording on “i” really confused me here, because you refer to these as “site i” but these are really indexes for site x visit. (?)  

Line 152: Missing R citation

Line 152: I suggest single quotes around the package name to avoid confusion.

Figure 4 caption: Mis-spelling at “realized abundances are assumed to be population sizes”

Figure 4 caption: need a comma “site-specific population estimates, which are…”

Line 215: Mispelled “complementary:

Author Response

See the PDF attached.

Reviewer 2 Report

Line 39 – 40  - I have no clue what the authors are trying to say in this sentence.

Line 92 – redundant start from previous paragraph - revise

Missing from methods – what was the mode of transportation to sampling sites – boat, car, did the boat have a motor, if so was a motor boat used during each survey.

Why was site visit or rather sampling time not included in the analysis? Terrapins show seasonal shifts in behavior and habitat use that suggest that time of year, sampling time, should be included in the model.

There is no description of how wind speed was measured.

There are no citations in the Statistical Analysis and given the detail of the model description it seems that citations are needed.

Line 155 – redundant already stated in methods

Line 180 – I think you mean detection rate because you actually do not know if terrapins were present or not when they are absent. Therefore I think it is okay to say that more turtles were detected in unexposed locations suggesting their abundance is higher there.

Figures 2 and 3 are redundant – I prefer figure 3 for these data.

Figure 4 once again in the legend should be detections and what is the x-axis. It is not labeled and not described in the legend.

Line 217 – I liked the way you stated this in your conclusion better where you discussed regional abundances.  I would assume that all the terrapins in Wellfleet Bay constitute a single population – see more comments below.

Line 248 – is and are

Paragraph beginning on 250 – I would guess that the population in Wellfleet Bay is a single panmictic population and therefore the comments in this paragraph suggesting that each sampling location represents a population is not valid. I think that the data speak more to habitat selection and presence absence of individuals, perhaps even to the extent that making statements about population size should include all the data. Clearly some animals move between some of these sites which means that they are likely to traverse exposed sites but linger in unexposed – perhaps feeding or mating.

Line 266 – sheltered and exposed shorelines occur naturally and may be present independentof any shoreline alteration – not sure what you are implying here  you do not identify if your exposed areas are due to human alteration e.g. hardening.

I am really surprised that Isdell, R. E., R. M. Chambers, D. M. Bilkovic, and M Leu. 2015. Effects of terrestrial-aquatic connectivity on an estuarien turtle. Diversity and Distributions 21:643-653. Was not cited in this study

Author Response

See the PDF attached.

Reviewer 3 Report

Title: Visual head counts: a promising method for efficient monitoring of diamondback terrapins.

Overview:

Diamondback terrapins (DBT) are a species of conservation concern in the eastern USA, and estimating range-wide abundance (and the drivers of regional variation in density) is a critical first step in making effective conservation decisions for this species. However, generating abundance estimates from traditional sampling methods (e.g., capture-mark-recapture) has proven challenging due to low capture efficiency. The authors hypothesize that repeated visual head counts (scans of the water surface for emerging terrapin heads), coupled with an N-mixture modeling framework, could provide a more efficient system for regional monitoring of DBT populations. The authors used this new head-counting system to estimate the monthly local abundance at 38 sites in Wellfleet bay, MA, and to assess the environmental drivers of abundance and detection probability in this system. The results showed that detection probability was negatively affected by high wind speed and overcast conditions, and that abundance was higher in unexposed vs. exposed sites. The authors conclude that visual head counts are an effective and efficient method for surveying DBT populations.  

General comments:

Overall, I was not convinced by the author’s claims about the utility of the proposed visual-head-count monitoring system. Every reasonable hypothesis should have a chance of getting rejected after the evidence is in. The way this manuscript is currently written, I don’t see exactly how the main hypothesis (that visual head counts are demonstrably useful for monitoring DBT populations) could be rejected (i.e., the argument as currently developed seemed tautological). Statistical models will generally run and produce results if enough data are available, so the more important question isn’t “will the model produce results?” but “are the results valid or useful?”. However, without independent validation data or model-adequacy tests it is difficult for the reader to assess whether the results are valid or useful. For this research to contribute to the scientific literature and DBT conservation, I think the authors need to present the results of one or more goodness-of-fit tests, in which they demonstrate that their observations could reasonably be produced from their fitted model. Secondly, in an ideal world the authors would show that their abundance/density estimates matched with an estimate derived from a more standard approach (e.g., CMR, distance sampling). Are any CMR data available at Wellfleet bay during similar time periods that could be used to help validate the results? If not, are there any other types of independent data that could be used to help validate the results?   

Since there are several hints in this abstract and ms that this method could be used to generate regional or range-wide abundance estimates, it would be useful if there were more details provided on how this might be done. How would you estimate total abundance with this framework? Also, which (if any) caveats should practitioners be mindful of if they were to implement this approach to estimate regional abundances/densities? I assume you would need to convert local abundance estimates into density estimates and then integrate these (spatially varying) density estimates across large areas? In any case, I think the ms could be strengthened with more information about how the results could scale up to make regional or range-wide predictions.

More information on the surfacing behavior in DBT (the process which exposes DBTs to detection in a visual head-count survey) would help to evaluate the adequacy of the sampling data for fitting the specified model. One of the assumptions that inherent to the N-mixture modeling framework applied here is that each of the 5 visual scans in each head-count survey represents an independent sample from the population. To evaluate whether this is an appropriate assumption for this system, I would like to see more information provided on how often DBTs come to the surface and (once they are at the surface) how long they tend to stay there.

Finally, one of the key results was that abundance estimates were higher for unexposed vs exposed sites. However, the detection probability model indicated that high winds substantially reduced detection probabilities due to wave action making it difficult to see turtle heads. I could be completely wrong here, but I might expect that exposed sites might have more wave choppiness in general. Did you try including exposed/unexposed as a covariate in the detection model? Is it possible that the increased abundance at unexposed sites is due to increased detection efficiency (lower wave chop) in these less exposed habitats? Perhaps just some additional clarification would help to allay this concern.   

Specific comments:

17 and elsewhere: why do you refer to this modeling framework as “so called” N-mixture model? I don’t see the point of the “so called” qualifier.     

83: is there any chance of double-counting individuals during a scan (e.g., they swim and resurface somewhere else during a single scan)? If so, hopefully that would happen so infrequently as not to affect the model fitting results. It would be nice to have a sentence somewhere that explains why double-counting within a scan is unlikely to occur.   

101,113: It seems strange to me that re-surveys performed at the same point locations in different months are treated as entirely separate abundance estimates. Is there reason to suspect that resurveys at 1 month intervals might not be completely independent? If so, could you include site as a random effect to account for this effect? Also, if site abundance was indeed highly variable over short time frames, how would you propose to use this method to estimate regional abundances, or regional abundance trends among years?  

102: If the sampled area could be defined, would it be appropriate to interpret population densities as an important model output? If so, it might be useful to demonstrate how this could be done, as well as any notable caveats to doing so.

131: The Poisson distribution, defined by only one parameter, is quite constrained and often is not an adequate descriptor of ecological processes.  Which is not to say I think you should avoid using the Poisson distribution, but it makes it even more important to test the model adequacy (goodness of fit). If the goodness of fit fails, then other, less constrained distributions (e.g., mixture models like neg binomial, hurdle models, etc.) are available.

140-142: Why did you only consider a single environmental covariate on abundance- are there any other hypothesized drivers of abundance in this system that you might include?

Author Response

See the PDF attached.

Reviewer 4 Report

Brief Summary

The aim of this pilot study was to demonstrate the usefulness of visual head counts combined with N-mixture models to rapidly assess terrapin population estimates. The authors found that variation in detection probability could best be explained by a combination of wind and temperature and that time of year and amount of exposure best explained terrapin abundance patterns. The authors do, however, admit that exposure is being used as a “catch-all proxy” for other covariates that are actually driving DBT abundance patterns, which is a major assumption that needs further explanation and support.

Broad Comments

                The authors do a good job of describing the visual survey methods and the model under consideration. I would, however, imagine that exposure could have an equally large effect on detection as on occurrence, so I would have liked to have seen how exposure influenced detection. If an area was more exposed, wouldn’t terrapins have been easier to see? The authors do state that some sites were discarded due solely to a lack of open water, so it is evident that exposure level and detection are related. Additionally, I did not see the length of time of each scan reported in the methods. I see the area surveyed and time between scans, but no mention of scan time. If scan times were consistent between sites, then I would expect scans at more exposed sites to be more efficient because it would be easier to spot terrapins at those sites, but if scan times varied based on exposure, this should have been included as a covariate in the detection model.

                I am impressed by the number of models explored and think the authors did a very thorough job in examining the role of each covariate under consideration in explaining terrapin detection and abundance patterns. I do, however, wish that more covariates were explored. The authors explicitly state that this was a pilot study and they plan on including more covariates in the future, so the question immediately becomes: was the study presented in this paper enough, as is, to add value to the field. This question is difficult to answer as the model had no validation. I am aware that model validation (like cross-validation) is rare in studies using Bayesian models, but it would demonstrate the validity of the method, which was not done here. Additionally, the only mention of how well the model fit the data was in the Appendix. I think that the supplementary information in the Appendix should either be supplementary, i.e. not necessary to fully understand the information presented in the paper, or should be included as part of the paper itself. This is especially true for Figure A2, which is the only evidence of goodness of fit given in the paper. It is important, and not supplemental, to note that the best model fit the data well.

                I do question why AIC was used as a metric of model comparison rather than DIC, which is the hierarchical modeling generalization of AIC typically used in Bayesian model selection. I would like to see either a justification of AIC in this case, or recommend using DIC instead.

                N-mixture models provide reliable estimates of abundance from repeated counts, so it is reasonable to apply count data generated from head-count surveys to this method. I would argue that the value of this paper is not so much in making the case that this can be done, but rather in making the case that it can be done reliably and produce reasonable estimates when the correct covariates are used. Saying the model can be built and produce results in general is not as interesting as producing a model that you can demonstrate does a good job. The authors need to either justify that exposure alone is capable of explaining terrapin abundance patterns, which they address briefly in lines 289-290, but fail to fully explain, or add more covariates that actually explain what is driving differences in terrapin abundance at different sites and through time.  

Specific Comments (in order of importance)

Line 164: This is an important note and potential flaw of the modeling framework, you cite the Discussion, but do not fully address the lack of closure in the Discussion.

Line 145: This topic needs to be broadly expanded – why did the authors feel comfortable making this assumption and what evidence supports this?

Lines 98-99: Not sure what is meant by “timing was assumed to be sufficient capture variation in the individuals available for detection” and not sure how citing the appendix figure adds anything. Please explain this in more detail. And the caption of Figure A1, is the last part of the last sentence a hyperlink? Or is there missing information? And in that caption (Lines 350-351) how is turnover related to variation between scans? The caption did not fully explain what information I was supposed to get from the figure. And there needs to be a legend explaining the color scheme.

Lines 113-115: Sentence did not make sense to me – recommend re-wording.

Lines 59-63: This sentence is poorly worded and lacks clarity.

Lines 85 and 86: There is no such thing as unsuitable habitat, if it is not suitable, it is not habitat.

Lines 191-193: Sentence did not make sense to me – recommend re-wording.

Table 1: p in Temp should not the italicized.

Round 2

Reviewer 1 Report

The revised manuscript is much improved, and the new figures are very helpful for understanding what has been done. The parametric bootstrap simulations combined with Figure 3 addressed my earlier concern over the way variation in lambdas were presented in the original manuscript's Figure 4.

I do strongly suggest one edit to Equation 4, which is to replace "Exposure_i" with the indicator variable "I[Exposed_i]" (with concomitant changes to the text in the paragraph above to replace Exposure with Exposed). In other words, the indicator variable represents a 0/1 switch between unexposed and exposed. As written now, Exposure_i reads like a continuous variable. Though this section of the methods is much clearer, this will further avoid confusion about the number of parameters to be estimated.

Author Response

Many thanks for reviewing the revised version of the manuscript. We understand the constraints on everyone's time, and very much appreciate your input on making this manuscript the best it can be. As such, we have dealt with your comments (see response) explicitly and in full.

[R1] The revised manuscript is much improved, and the new figures are very helpful for understanding what has been done. The parametric bootstrap simulations combined with Figure 3 addressed my earlier concern over the way variation in lambdas were presented in the original manuscript's Figure 4.

[response] Thank you, and thanks for the original comments related to goodness of fit and the clarity of the technical material. We are really happy with the way it turned out as a result.

 [R1] I do strongly suggest one edit to Equation 4, which is to replace "Exposure_i" with the indicator variable "I[Exposed_i]" (with concomitant changes to the text in the paragraph above to replace Exposure with Exposed). In other words, the indicator variable represents a 0/1 switch between unexposed and exposed. As written now, Exposure_i reads like a continuous variable. Though this section of the methods is much clearer, this will further avoid confusion about the number of parameters to be estimated.

[response] Another great point. In addition to changing exposure to exposed, in the text and equations, we have also included more explicit language being absolutely clear about it being a binary / two-class covariate (see lines 141-144, 144-146, Eq 4, 176, 194-195).

Reviewer 4 Report

I believe the authors have adequately addressed the original concerns I raised. I find this study to be well written and easy to understand, particularly after the clarifications made by the authors. I think this study demonstrates that the n-mixture model framework is applicable to DBT visual-survey datasets and I look forward to reading the results of future papers on this study system.

I have a couple of minor suggestions. While the authors do a good job of stating that this is preliminary, I think it would be beneficial to have a line or two in the last paragraph of the discussion explicitly stating that the abundance estimates generated in this study are likely not reliable due to the lack of explanatory covariates, just so these estimates are not used in future studies on DBT. (This is obvious given the dramatic change in the abundance estimates at exposed sites between the last version of this ms and this version (as in Figure 4); which is evidence that the estimates are likely to change dramatically as covariates are added/changed.) I know that the authors have stated this is a pilot study and the goal was to show that the method works, but since population estimates are reported, I think clarifying their accuracy explicitly is important.

Additionally, I think the goodness-of-fit test is a critical part of the authors argument that this model framework is useful for this dataset and the figure related to this test is highly relevant and should be included in the ms. So, to clarify, I think Figure A2 should be a figure in the paper itself, not just the supporting material. The table, I agree, is supplementary, given that the relevant numbers are listed in the text.

Author Response

See cover letter attached.